# Machine learning-based framework for wall-perching prediction of flying robot

Yunian Shen [1,2] ✉, Chenxi Mao[1,2], Zeyu Qi[1], Kunpeng Liu[1], Weixu Zhang[1] & An Cao[1]

In contrast to the diverse landing solutions that the animals have in nature, human-made aircraft struggle when it comes to perching on vertical walls. However, traditional dynamic simulations and experiments lack the high efficiency required to analyze the perching and design the robot. This paper develops an efficient machine learning framework to predict vertical-wall perching success for flying robots with spines, overcoming traditional methods' inefficiency. A validated knowledge-based model computes the robot's transient dynamics during high-speed perching, identifying key success factors. By training the mixed sample data, a data-driven model has been proposed to predict the success or failure of an arbitrary perching event. Here, we show that this high-precision prediction optimizes robot control and structural parameters, ensuring stable perching while drastically reducing the time and cost of conventional design approaches, advancing the flying robot capabilities.

Perching is an innate ability of animals, which means they can land and stay on a branch or another suitable surface. For example, flies can land upside down on a ceiling[1], gliding geckos crash into tree trunks with their tails to stabilize landing[2], and bats land on the tops of caves to rest[3]. Meanwhile, creatures have evolved a variety of methods for landing, perching, locomotion, and take-off to adapt to various irregular natural and artificial surfaces[4]. In contrast to animals, the perching approach of robots (especially aircraft) for surfaces is tougher, a flat ground is necessary. However, having these perching abilities similar to animals can help expand the use of the robots in extreme environments and improve their mobility, robustness, and working time[5,6]. For instance, in search-and-rescue operations, perching mechanisms would allow drones to land on walls or power lines, enabling persistent environmental monitoring from vantage points while conserving power[7].

Consequently, achieving reliable perching capability on non-horizontal surfaces has emerged as a significant research frontier in aerial robotics. The optimization challenges pertaining to pitch angle, velocity, and lift force during avian perching[8] are equally encountered in developing perching technologies for aerial robots. Zufferey and Tormo-Barbero[9] presented a process to autonomously land an ornithopter on a branch. This method explains the combined operation of a pitch-yaw-altitude flapping flight controller, an optical close-range correction system, and a bistable claw appendage design that can grasp a branch within 25 milliseconds and reopen. Guo, Tang et al.[10] developed an actuated modular landing gear system achieving stable perching and resting of UAVs on diverse structures. Roderick, Cutkosky, and Lentink[11] designed a biomimetic robot achieving dynamic perching on complex surfaces with concurrent grasping of irregular objects, demonstrating that closed-loop balance control critically expands viable perching parameter ranges. Hsiao, Bai et al.[12] presented a flying robot achieving perching on diverse substrates – including wet/dry overhangs and walls – via an ultralight attachment module constituting merely 3.3% of total mass. Liu, Tian et al.[13] proposed a contact-adaptable robot for perching featuring all-in-one electroadhesive pads, enabling ceiling surface attachment and significant onboard energy conservation. Liu, Dong et al.[14] developed a multifunctional aerial manipulation system using a composite cup structure (soft inner cup + rigid outer shell), enabling execution of perching or lateral aerial grasping tasks while reducing reliance on precise control through the soft cup's adaptability to multicopter-induced angular errors. While the aforementioned studies have made

[1]Department of Mechanics and Engineering Science, School of Physics, Nanjing University of Science and Technology, Nanjing, P. R. China. [2]These authors contributed equally: Yunian Shen, Chenxi Mao. ✉e-mail: yunianshen@njust.edu.cn

substantial progress in developing and characterizing novel perching mechanisms, no research to date has established predictive frameworks for perching. Nevertheless, perching—which involves rapid maneuvers and is subject to strict velocity constraints—remains one of the most critical challenges for flying robots. This is largely because the inability to predict unsuccessful landings on vertical surfaces is highly likely to result in severe damage to the robot[15]. Establishing a predictive framework for perching would therefore greatly enhance the success rate of wall-perching maneuvers and reduce the losses associated with landing failures.

To predict the wall-perching effect and design the structure configuration of the flying robot with spines, people have recourse to traditional dynamic simulation[16] and experimental method[14] at present. However, traditional dynamic simulations lack the high efficiency required to handle the substantial computational workload. Furthermore, the limited simulation results are difficult to form the practical control strategies for perching. Similarly, the observation of a limited amount of experimental data from perching experiments is insufficient for the design of flying robots. Fortunately, machine learning, as a part of Artificial Intelligence (AI) technology, offers a promising solution for providing robots with perching experience and improving their capabilities. This potential will be realized through advanced ML implementations, as demonstrated by recent breakthroughs in flight control domains.

The integration of machine learning (ML) into flight systems to enhance autonomous capabilities has reached a sophisticated level of maturity such as Kaufmann, Bauersfeld et al.[17] presented a champion-level autonomous racing system employing deep reinforcement learning fusing simulated training with physical vehicle telemetry. Ouahouah et al.[18] proposed an obstacle avoidance algorithm for UAVs based on probability and Deep Reinforcement Learning (DRL). Recently, ML-driven perching control studies for horizontal surfaces or non-horizontal surfaces are constituting a rapidly evolving frontier. Waldock, Greatwood et al.[19] implemented a DRL framework integrating a nonlinear constraint optimizer (IPOPT) with a DQN agent to compute perching trajectories for horizontal surfaces. This approach derives optimal pre-contact pitch/velocity parameters that guarantee structurally stable perching on horizontal surfaces. de Croon, De Wagter et al.[20] proposed a machine learning method enabling flying robots to acquire optical-flow-based strategies for complex tasks such as landing, resulting in faster, smoother landings. Ladosz et al.[21] employed visual sensing and DRL, incorporating a Deep Regularized Q-learning algorithm and a custom-designed reward scheme to enable autonomous landing of UAV systems on moving platforms. Habas and Cheng[15] replicated fly landing behaviors in micro-quadcopters using a generalizable control policy for arbitrary ceiling-approach conditions. Their framework employs reinforcement learning in simulation to optimize discrete sensorimotor pairs across diverse approach velocities/directions, with the first-stage "Flip-Trigger" control implementing a one-class support vector machine (SVM). Despite these advances, no existing ML addresses the challenge of predicting wall-perching outcome while co-designing robot structures – a critical capability for designing flying robotics in complex environments.

In this work, to bridge this gap, a machine learning-based framework for the wall-perching prediction and design of the flying robot with spines is developed in this paper. The framework synthesizes the machine learning, computational impact dynamics and experiment. Firstly, an accurate knowledge-based model, validated for accuracy through experimentation, has been developed to calculate the transient responses of the flying robot during perching. The key factors that influence the success rate of perching at relatively low landing speeds are identified. Then, a flying robot prototype is constructed to conduct the perching experiment. In addition, the mixed sample datasets are obtained, comprising experimental data from perching experiments and simulation data computed using the knowledge-based model. By training the mixed sample datasets, a data-driven model is established to predict the success or failure of an arbitrary perching event. The results of this study can aid in designing flying robots and preventing perching failure.

## Results

### Robot's structure and perching strategy

To achieve the landing and climbing functions, a spiny flying and wall-climbing robot is designed based on the multimodal robotic design philosophy[5]. The whole robot is composed of three subsystems, namely the flight system, landing system, and attachment system (Fig. 1a). Here, the flight system mainly consists of a micro quadcopter, which can provide the robot with flying capability. The landing system consists of two flexible tail structures connected by a transverse beam with two wheels (Fig. 1a). During landing, the tail will make initial contact with the vertical wall. The huge impact force is reduced thanks to the large structure compliance of the tail we designed. The attachment system is designed with two carbon fiber rods with spines (Supplementary Figs. 1, 2). The spine morphology exhibits structural similarity to beetle tarsal claws[15]. Due to the very tiny tip of the spine, it can crawl stably on the wall.

The robot achieves perching through four designed stages (Fig. 1b-e). In the first stage, the robot approaches the vertical wall using its flight system. Then, the second stage is a contact-impact phase, where the robot uses its flexible tail rods to impact the wall. Subsequently, it enters the third stage (i.e., rotation stage). The robot's wheel contacts the wall, and then the entire robot body rotates upwards around the wheel. In the fourth stage, the robot uses spines in its attachment system to climb on the wall, and the whole perching process is completed. Although the attachment system (Fig. 1a) can help the robot climb on the wall, this paper focuses on the landing and perching problems of the robot. For detailed descriptions of the spine structure and wall-climbing mechanism, please refer to the Supplementary Note 1.

### Numerical evaluation for perching based on knowledge-driven model

To analyze the landing and perching behavior of the robot, humans resort to a contact-impact dynamic model naturally. This model is a typical knowledge-driven model based on the principles of impact dynamics of a flexible multibody system. In this study, the model is discretized using the finite element method (FEM) (Fig. 2a), and calculations are performed in the LS-DYNA software. We use this model to evaluate the success or failure of perching in different maneuvering states before landing. The numerical results guide the experiments and are used to build the subsequent data-driven model, which will be discussed the next section.

The numerical simulations for 114 cases of maneuver states before landing have been performed. The alterable parameters of maneuver states include initial pitch of the flying robot $\theta_0$, the initial horizontal incidence velocity $v_0$ and the lift $F_L$. Figure 2b illustrates the displacement of the spine after the tail contacts the wall, which was calculated based on the knowledge-driven model. The peaks of the dashed curves represent the quick release of deformation energy from the tail. In Fig. 2b, when the positive direction of the $X$-axis rotates counterclockwise to align with the robot's axis, $\theta_0$ is defined as positive. Conversely, when the rotation is in the opposite direction, $\theta_0$ is considered negative. The other attitude angles and flying velocities are all equal to zero. It should be noted that the lift $F_L$ is an active force, it will be controlled during the contact-impact process. It is set as an increasing or decreasing linear function of time $t$ (Fig. 2c, d) to analyze the effect of $F_L$ on the success of landing. The gravity is a constant value during the whole landing process.

By our computation for 114 cases based on the knowledge-driven model, it can be found that there are 4 types of typical landing behaviors (Fig. 2e). And regardless of what the initial conditions are, the

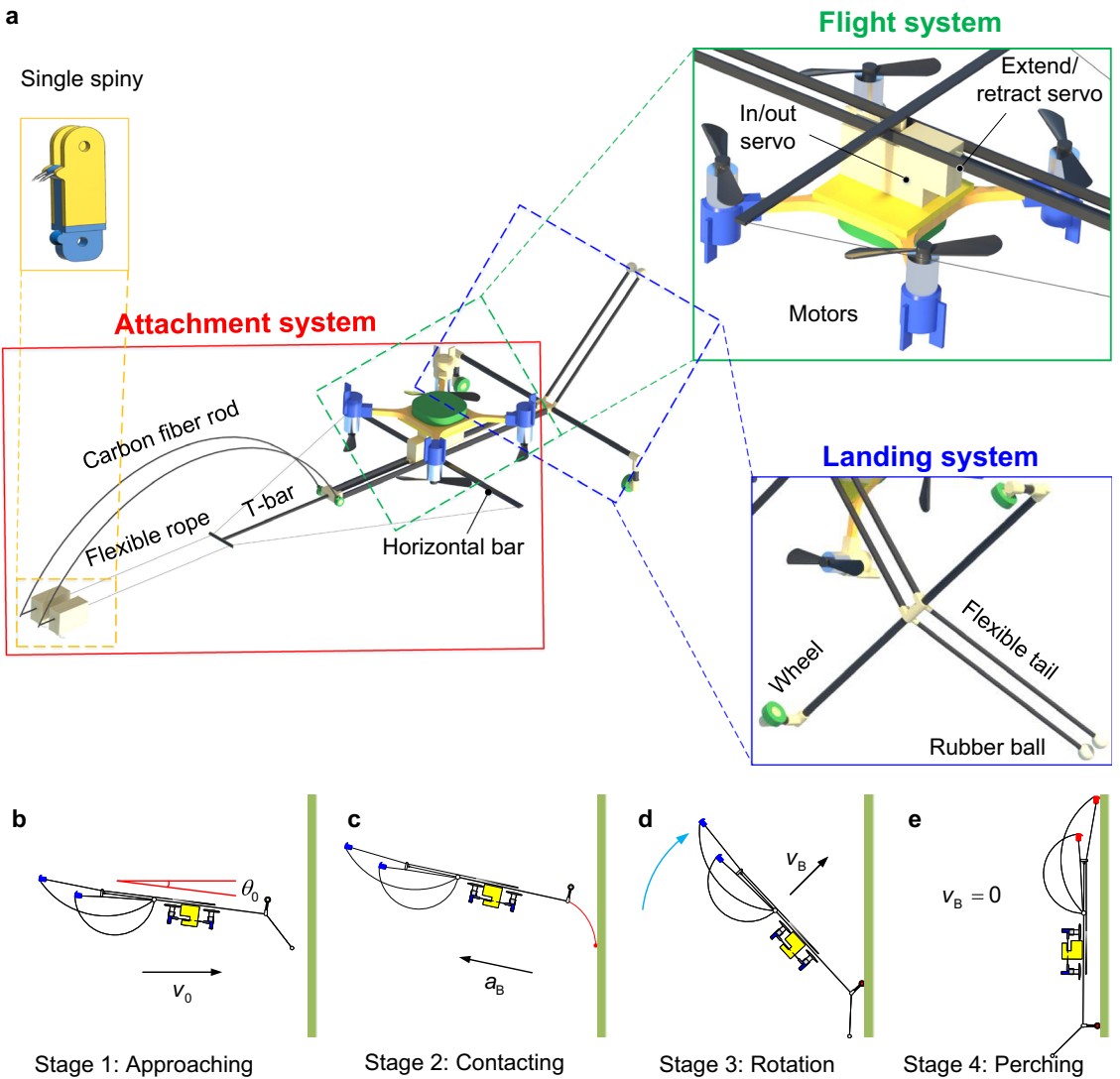

**Fig. 1 | Landing process of robot and its structure composition. a** The structure composition of the robot including attachment, flight and landing systems. **b** Robot flies approaching wall, where $v_0$ and $\theta_0$ are the initial horizontal incidence velocity and initial pitch of flying robot, respectively. **c** The robot contacts the wall by its tail, where $\alpha B$ is the robot's body acceleration. **d** The robot executes an upward flip action, where $vB$ is the velocity of the mass center of the robot. **e** The robot grasps and perches the wall.

resultant velocities of spine always experience multiple oscillations since the tail occurs obvious vibration under contact-impact. Figure 2e also shows the envelop regions that are summarized from 114 landing events corresponding to the 114 data points in section "Data-driven model for perching". The green area means the robot landed successfully while red and purple areas mean robot failed to land (LR means lift receding and LS means lift strengthening). Meanwhile, some simultaneous robot postures and stress states during the landing are also given. From Fig. 2e the 4 types of typical landing behaviors can be summarized as follows:

1) Perfect landing (success). The characteristic feature of such landing behavior is that the robot's mass center trajectory approximates a semicircular path, with negligible body sway after attachment. The corresponding mass center velocity is indicated by the black curve in Fig. 2e. To get these results, the robot must have the most ideal landing state before contacting the wall. i.e., the robot almost flies horizontally, ($-8° < \theta_0 < -5°$) and the sum of velocity of the mass center and the pitch are mid-value ($0.6$ m/s $< \theta_0 < 0.8$ m/s). Then, the tail contacts the surface (Fig. 2f) and it occurs bending deformation. Then, the robot performs a fixed-axis rotation where the rotation center is the contact point of the

tail. The tail does not keep in contact with the wall until the wheels contact with the wall. Then, the robot body flips upwards (Fig. 2g) using the wheels as the rotation center. Finally, the robot's spines grasp the surface and attach successfully.

2) Normal landing (success). The characteristic feature of such landing behavior is that the robot's mass center trajectory approximates a horseshoe-shaped path, with significant body sway after attachment. The corresponding mass center velocity is indicated by the blue curve in Fig. 2e. To achieve this landing process, the robot requires the initial states of $-5° < \theta_0 < -2°$ for the angle and $0.8$ m/s $< v_0 < 1.0$ m/s for the velocity before contacting the wall. Subsequently, the robot performs a plane motion to achieve upward flips. However, the wheel undergoes a transition between contact and separation with the wall (Fig. 2h) during the entire upward motion. Finally, the wheel will contact the wall again at the end of the landing process. In other words, the wheel makes contact twice during the landing process.

3) Low-kinetic energy landing (failure). This typical behavior and landing process is shown as the red curve in Fig. 2e. The initial conditions for this landing behavior are $v_0 < 0.4$ m/s and $\theta_0 > -4°$. Since the initial kinetic energy is low, the robot cannot perform an

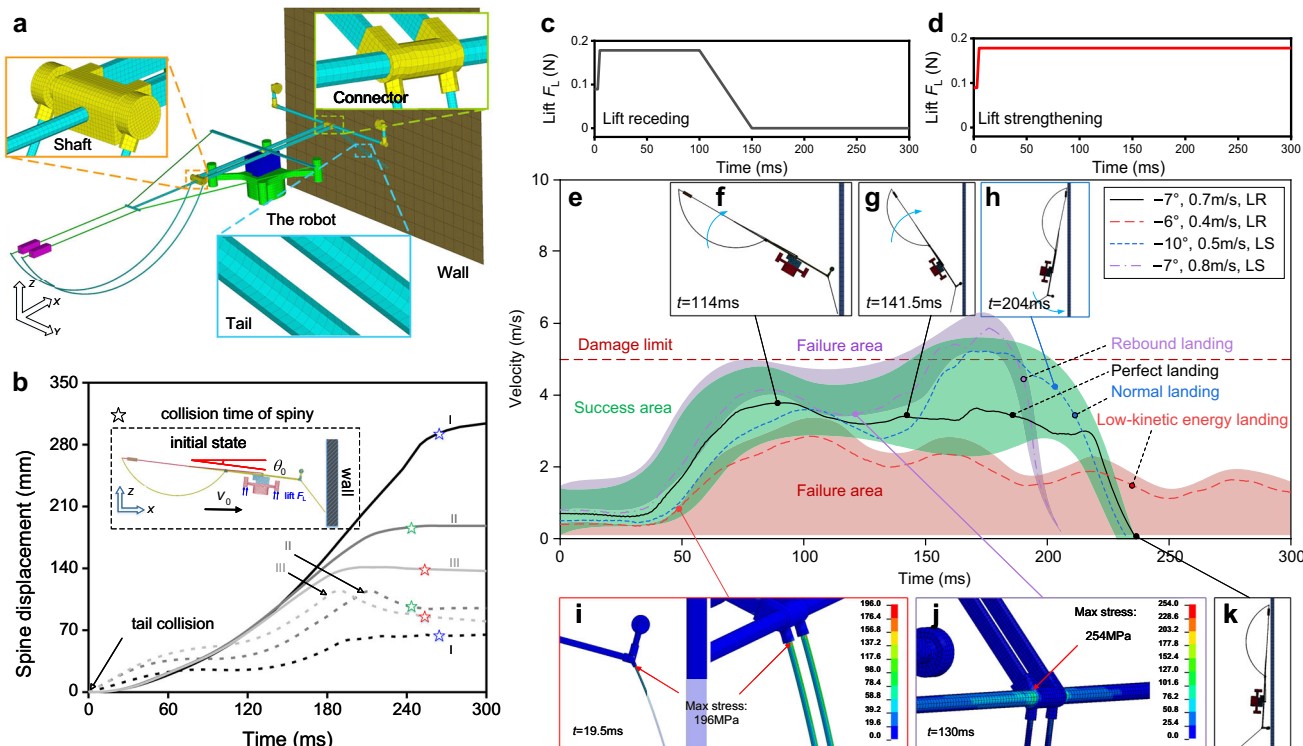

**Fig. 2 | Knowledge-driven model of the robot and its numerical simulation.**
**a** Discrete model based on FEM, which is used to solve the contact-impact problem during the landing event. **b** Displacement of the spine under three sets of initial landing conditions with lift strengthening (I indicates $\theta_0=-5°$ and $v_0=0.4$ m/s; II indicates $\theta_0=-5°$ and $v_0=0.8$ m/s; III indicates $\theta_0=-10°$ and $v_0=0.8$ m/s). The solid and short dash lines are the displacements in the $x$ and $z$ direction, respectively. The symbol star represents the occurrence of collision between the spine and wall. **c** The lift curves during the landing event for receding cases. **d** The lift curves during the landing event for strengthening cases. **e** The resultant velocity of the spine during the landing process for 4 typical landing behaviors. The success and failure envelop regions (i.e., the green and red areas) that we summarized for the landing events under 114 sets of initial landing conditions are also shown. The three values

in legend in this figure are pitch of flying robot $\theta_0$, initial horizontal incidence velocity $v_0$ and lift $F_L$, respectively (where LR means lift receding and LS means lift strengthening). **f** The robot undergoes a fixed-axis rotation where the rotation center is the contact point of the tail during the "perfect landing" process. **g** The robot body flips upward using the wheels as the rotation center during the "perfect landing" process. **h** The wheel separates from the wall and then contacts it again during the "normal landing" process. **i** The maximum von-Mises stress appears at the root of the landing rod when the robot contacts the wall. **j** The maximum von-Mises stress appears in the connecting piece. If the high velocity causes excessive stress, the structure will be damaged. **k** Configuration of robot when it is in the success perching state on the wall.

upward flip motion, despite the fact that the lift of the quadrotor aircraft is sufficient and its tail does not rebound from the wall. The robot will execute an upward straight motion parallel to the wall surface.

4) Rebound landing (failure). This landing behavior has two forms of rebound landing. The first is when the robot achieves an upward flip motion, but the spine rebounds when it contacts the wall due to the large contact force at the spine. This typical behavior and landing process are shown by the purple curve in Fig. 2e. In this case, the initial conditions are generally $v_0>1.0$ m/s and $\theta_0<-10°$. The failure is caused by the spine's inability to grasp the wall tightly. The second form is when the robot directly rebounds as the tail contacts the wall, preventing it from achieving an upward flip motion. In this case, the initial conditions are generally $v_0>1.0$ m/s and $\theta_0>-2°$. The occurrence probability of this form is very low because $\theta_0$ is generally less than $-2°$ during the actual flight of the robot under artificial operation. Besides, Fig. 2i–j shows the stress concentration phenomenon in the structure of the landing system. The critical areas where structural failure can occur are primarily located in the root zone of the tail.

Overall, the conditions such as initial velocity, initial pitch, and lift of the robot have very complicated effects on landing behavior and the landing process. They sometimes work together to achieve successful landings (Fig. 2k), but other times they conflict with each other and

lead to landing failures. It is more intuitive to understand that if the velocity is slower and the absolute value of the pitch angle is smaller, lift assistance is required. On the other hand, if the velocity is faster and the absolute value of the pitch angle is greater, increasing lift may cause the robot to rebound from the wall surface. However, these relationships are not absolute. Therefore, one of the greatest challenges in landing on a vertical wall is predicting success or failure.

Through our calculations, it has been determined that the time required for each computation using the knowledge-driven model is substantial. Moreover, it is impractical to exhaustively explore all working conditions and control parameter combinations using this method. On a computer equipped with an Intel CPU 6700 K and DDR4 16GB RAM, the cost of each landing process generally requires around 48 h. Thus, there is an urgent need for an efficient and accurate calculation model to enable numerical evaluation of all landing conditions. Fortunately, the emergence of machine learning methods has made this achievable. A comprehensive discussion on this topic will be presented in the section titled "Data-driven model for perching".

## Experimental data for perching
In the experiment of landing on the wall, the robot firstly takes off from a horizontal surface and then is operated randomly to land on the wall, with different initial pitch of flying robot $\theta_0$, initial horizontal incidence velocity $v_0$ and lift $F_L$. The video of the experiments were recorded to obtain experimental data (Supplementary Figs. 3, 4). For more details

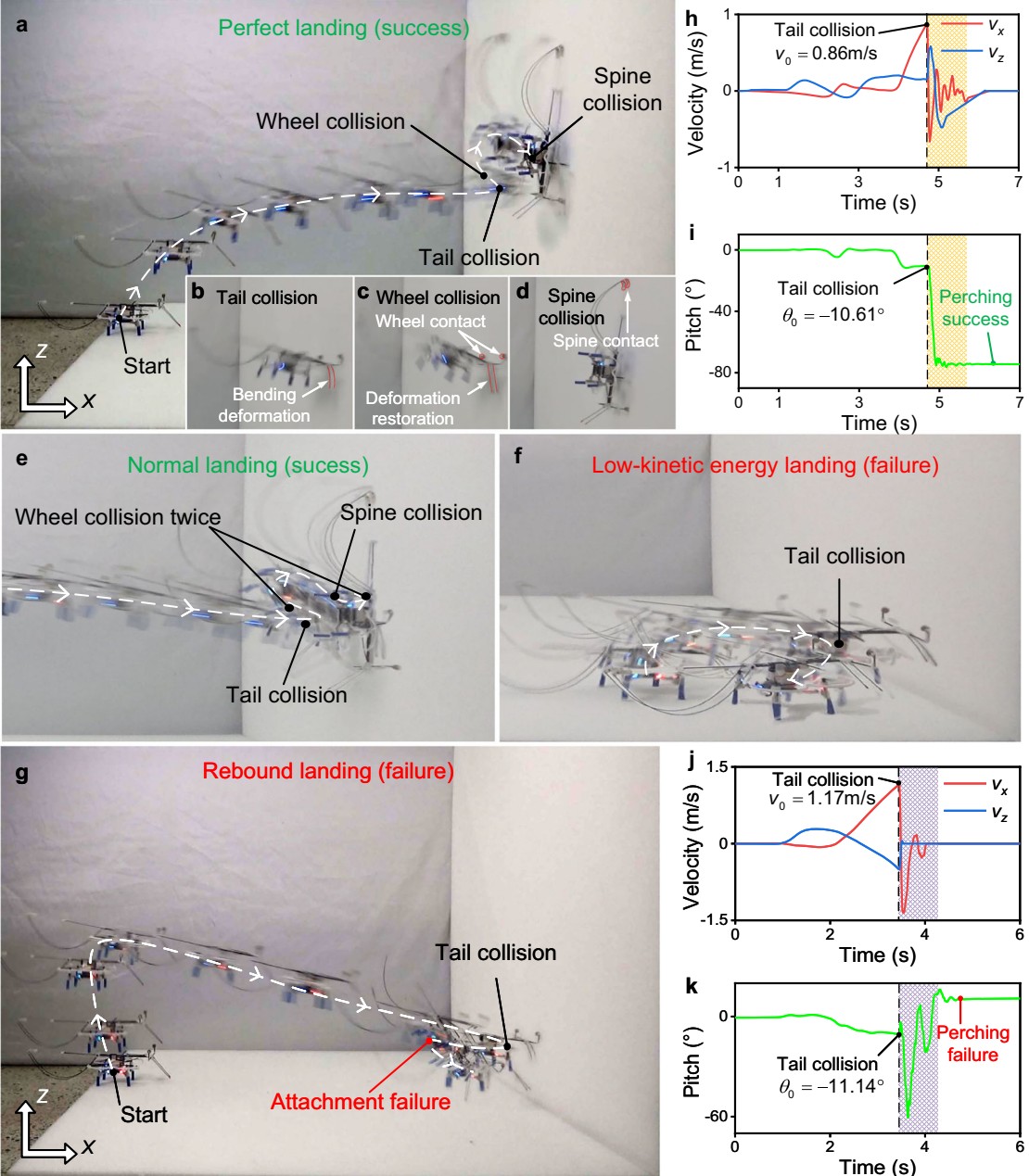

**Fig. 3 | Experimental image and sensor data for the robot's landing. a** The motion track for the perfect landing (success). **b** The tail contacts the surface during the perfect landing (success). **c** The wheels contact with the wall during the perfect landing (success). **d** The robot's spines grasp the surface and attach during the perfect landing (success). **e** The motion track for the normal landing (success). **f** The motion track for the low-kinetic energy landing (failure). **g** The motion track for the rebound landing (failure). **h** The velocities of the robot's mass center during the perfect landing (success). The shadow area indicates the robot is transitioning from a tail collision to a spine collision. **i** The robot's pitches during the perfect landing process. **j** The velocities of the robot's mass center during the rebound landing (failure). **k** The pitches of robot during the rebound landing (failure).

about the process of experiment, refer to Supplementary Note 2. The parameters, including $\theta_0$, $v_0$ and $F_L$, were intentionally varied before colliding with the wall to provide a diverse range of experimental results for machine learning in subsequent chapters. This experimental data serves two main purposes: firstly, it can be used to validate the accuracy of the computational results obtained from the knowledge-driven model. Additionally, it can be utilized to identify general patterns in the landing behavior by observing the robot's actual landing event.

Figure 3 shows the experimental images of 4 typical landing behaviors. The video of the continuous movement of the flying robot can be viewed in the Supplementary Movie 1. The movement

trajectories for perfect landing (success), normal landing (success), low-kinetic energy landing (failure), and rebound landing (failure) are depicted in Fig. 3a–g, respectively. It has been observed that the detailed characteristics captured by the experiment are consistent with those summarized by the numerical evaluation. Figure 3h–k represents the velocities of the mass center and the pitches of the robot during the perfect landing and rebound landing processes, respectively. During the perfect landing, the pitches change linearly to a constant value, while in the rebound landing, the pitches change nonlinearly from negative to positive.

The control parameters have a great influence on the landing responses. Due to its high cost and time consumption, merely a limited

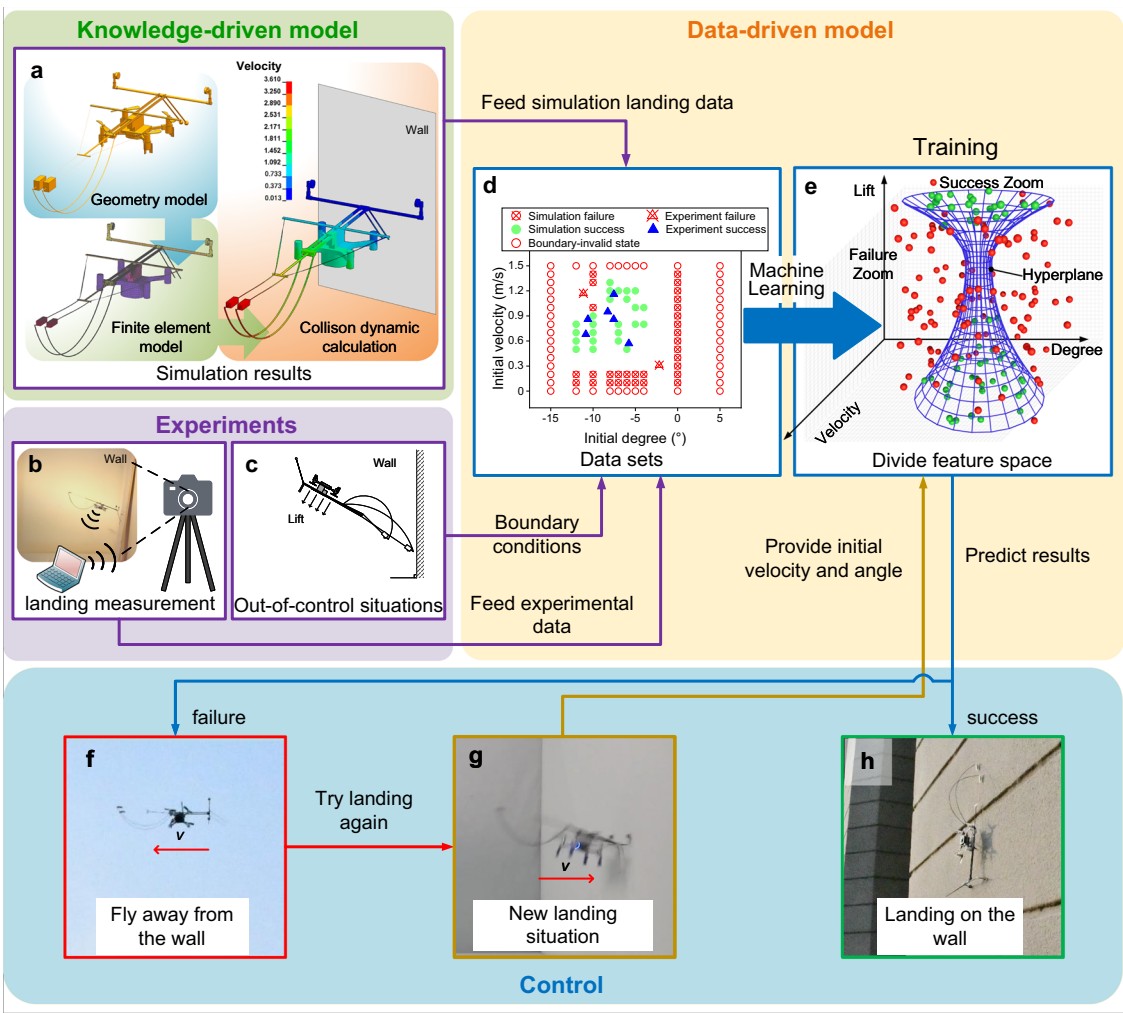

**Fig. 4 | The establishment of data-driven model, and how to use it to predict perching outcome. a** Simulation for robot landing based on the knowledge-driven model. **b** Actual experiment for robot landing. **c** Extreme state when the robot is out of control. **d** Data points obtained from simulations based on the knowledge-driven model and actual experiments under different lift forces. **e** Data-driven model of landing behavior is established after training using machine learning. **f** If the data-driven model predicts a failure in landing, the robot will abort the landing, fly away from the wall, and prepare for another landing attempt. **g** The robot perform a new landing. When it approaches the landing surface again, predictions are conducted once more through the data-driven model. **h** The robot lands on the wall if the data-driven model predicts a successful landing.

number of landing experiments can be conducted, and the experimental data under a limited set of control parameters can be recorded. It is hard to conduct experiments covering all working conditions. Nevertheless, the limited experimental results are still valuable, which can be used as sample data for establishing the data-driven model in the next section.

## Data-driven model for perching

Generally, "learning for simulation" is a traditional idea in AI fields. Recently, some scientists have hypothesized that "simulation for learning" may be one trend in computer graphic science[22]. Here, based on mixed samples including computational data obtained from a knowledge-driven model and experimental data, we have established a data-driven model to predict the landing of a robot. In this work, we propose a technique that combines simulations based on knowledge-driven models and machine learning based on data-driven models, in order to leverage the best aspects of both.

The entire training and prediction process is shown in Fig. 4, and includes the following 5 steps:

Step 1: Preliminary structural design of the robot. The preliminary design of the robot and its landing strategy are based on Newton mechanics theory.

Step 2: Achievement of data sets. An accurate three-dimensional finite element model (i.e., knowledge-driven model) is established to simulate the landing of the robot (Fig. 4a). The computation work of the contact-impact dynamic responses is performed in the software LS-DYNA. Through the computation for 114 cases of maneuvering states, the landing results are obtained. The actual landing experiments with the robot prototypes have also been conducted (Fig. 4b). To define the boundaries of our dataset (Fig. 4c), we identify data points corresponding to invalid states, including out-of-control events. This dataset is composed of the robot's pitch degree, velocity, lift data, and success/failure outcomes extracted from both computational and experimental results (Fig. 4d and Supplementary Data 1).

Step 3: Establishment of the data-driven model. To ensure the machine learning framework achieves effective landing predictions for different lift modes, the dataset is divided into two parts based on lift receding and lift strengthening. Each part contains two features: initial horizontal velocity and initial angle. The dataset is then distributed within a three-dimensional feature space composed of initial horizontal velocity, initial angle, and lift mode (Fig. 4e), representing landing outcomes under various initial conditions. Additionally, there is no noise in the dataset, hence the noise reduction is not required.

Step 4: Training. After training with the machine learning models, a decision boundary for Multi-layer Perceptron (MLP) and Random Forest (RF) (i.e. hyperplane for Support Vector Machine) is generated in the landing feature space under different lift conditions (Fig. 4e). The data points representing successful landings are separated from those representing failed landings by the decision boundary. The farther a data point is from the boundary within the decision boundary, the higher the probability of a successful prediction by the model, and the greater the confidence of the robot in a successful landing. By adjusting the model parameters that affect the construction of the decision boundary, we can obtain the optimal model with the highest confidence in its predictions.

Step 5: Prediction. According to the training results, when the robot is in a landing situation, the success or failure of landing will be predicted by the data-driven model, and the subsequent behavior of the robot will be determined. In the future actual application of perching, if the prediction result computed by onboard computer indicates failure, the two motors near the wall accelerate their rotational velocity to cause the robot to decelerate, then the robot will adjust its flight states in real-time to fly away from the wall (Fig. 4f) and seek a new landing opportunity (Fig. 4g). Conversely, if the prediction result indicates success, the two motors located away from the wall accelerate their rotational velocity to make the robot flip upwards, and the robot will directly land on the wall (Fig. 4h).

The entire computational process is following: After the primary selection of data collected from the robot's sensors, certain influential factors that affect the success or failure of landing can be determined manually. These influential factors can be employed as dimensions within the dataset. These influences can be interconnected with each other, and there is no limit to the number of dimensions. Both simulated and experimental data samples are utilized to construct datasets. Generally, a larger dataset will yield more accurate prediction results. However, if the selected range of the sample data can encompass all various scenarios and its distribution in the state space is more uniform, even a smaller dataset can still achieve desirable outcomes. The proportion of simulated and experimental data can be adjusted after estimating the time and cost considerations. Generally, simulated data is less expensive and more efficient. When the robot encounters a new scenario, we can instantly predict the outcome, allowing the robot to respond accordingly based on the prediction result. Ultimately, this enables the robot to intelligently handle various new challenges related to perching.

This paper employed the Python language to develop the machine learning program. In this study, the datasets were obtained under proper surface conditions, which are sufficient for the robot to land easily. For soft surfaces with low rugosity, the proper condition is when the surface Brinell hardness is less than 2. Many tree bark surfaces can satisfy this condition. For hard wall surfaces, a high level of rugosity is required. Our tests indicate that the surface static coefficient of friction $\mu$ approximately needs to be greater than 0.2, and there should be at least a 20% difference between static and dynamic friction.

After preparing the dataset, our goal is to ensure that the machine learning method achieves high prediction accuracy. Therefore, we need to optimize the data-driven model. During the model training process, we use the same dataset and continuously adjust the parameters of the generated model in order to identify the best parameter configuration (for SVM and RF) that yields high prediction accuracy. In this way, we are able to find the optimal model. Specifically, the landing success region and failure region are distinguished by the decision boundary. The accuracy of the decision boundary is determined by the ratio of the correctly classified data to the total dataset. Be different from SVM and RF, MLP do not need to identify the best parameter configuration.

Figure 5 presents wall-perching results calculated using data-driven models based on MLP, SVM, and RF methods, respectively. Each model predicts outcomes across three dimensions: lift force, pitch angle, and initial velocity. The lift curves illustrated in Fig. 2c, d remain

employed here, the hybrid dataset (Fig. 5a, b) used for training combines simulation results from our knowledge-driven model with experimental data. Figure 5c–h displays prediction results from MLP, SVM and RF, respectively. These results demonstrate that regardless of the machine learning method, the decision boundary envelope encompasses a larger successful perching region when lift generation is present, indicating that providing lift enhances landing success rates for flying robots. Comparative analysis reveals considerable differences among the three methods' predictions, with MLP demonstrating relatively superior performance over SVM and RF. Detailed descriptions of the data-driven models and corresponding machine learning methods are provided in Supplementary Note 3.

## Closed-loop design analysis

The machine learning-based framework can not only predict the landing, but also guide the design of the flying robot. In the initial design phase, some basic premise need to be considered for achieve the successful landing and perching on the wall, including the following points: 1) The mass of robot should be as light as possible; 2) The structure where the robot collide with the wall needs to have a certain buffering capacity to reduce the impact load; 3) The landing structure is beneficial for the robot to flip up from the horizontal state to the vertical state after collision, so as to achieve successful grasping.

Therefore, in view of these three requirements, the overall structure of the robot was preliminarily designed according to the existing experience and knowledge. Compared with other structures, the landing rod is particularly critical for the successful landing of robots. In this paper, the landing rod structure was preliminarily designed as a flexible carbon fiber rod structure. Compared with other forms of buffer structure, its biggest advantage is that it has the lightest weight under the same size, and its structural form is the simplest, which is very conducive to the assembly of the structure. On the basis of the initial design, there are still many parameters to be determined, including the length of the landing rod $l_a$, the angle $\alpha$ between the landing rod and the horizontal structure of the robot, etc. These parameters directly determine the success or failure of landing. The length of the landing rod $l_a$ determines the bending flexibility of the structure and the cushioning performance of the structure; The included angle $\alpha$ determines whether the flying robot can flip up at a reasonable speed. Therefore, the detailed closed-loop optimization design of these key parameters will be carried out below.

The process of closed-loop optimization design is shown in Supplementary Fig. 5. During the detailed closed-loop optimization design phase, we focused on optimizing the landing rod structure of the robot while keeping other components unchanged. Building upon the initial design, we developed multiple landing rod configurations with varied parameter combinations. The $l_a$ was incrementally increased from 2 cm to 4 cm in 1 cm steps, while the $\alpha$ was expanded from 20° to 60° in 5° increments, generating numerous parameter permutations. To identify the optimal parameter combination, we employed an iterative approach. This involved comparing machine learning-predicted successful landing zones across different parameter sets until the resulting hyperplane region achieved maximum coverage in the two-dimensional feature space defined by the robot's initial horizontal velocity and initial angle—thus accomplishing closed-loop structural optimization. Each iteration integrated FEM simulation analysis, physical landing experiments and machine learning to solve the hyperplane zone in its parameter combination. Through this process, we determined the robot's optimal structural configuration and geometric dimensions (Supplementary Fig. 6 and Supplementary Table 1). Additionally, this study optimized the lift conditions during the robot's landing phase.

Additionally, for the robot designed in this study, the preceding section determined the optimal set of initial horizontal velocities and initial angles for landing by comparing successful landing regions

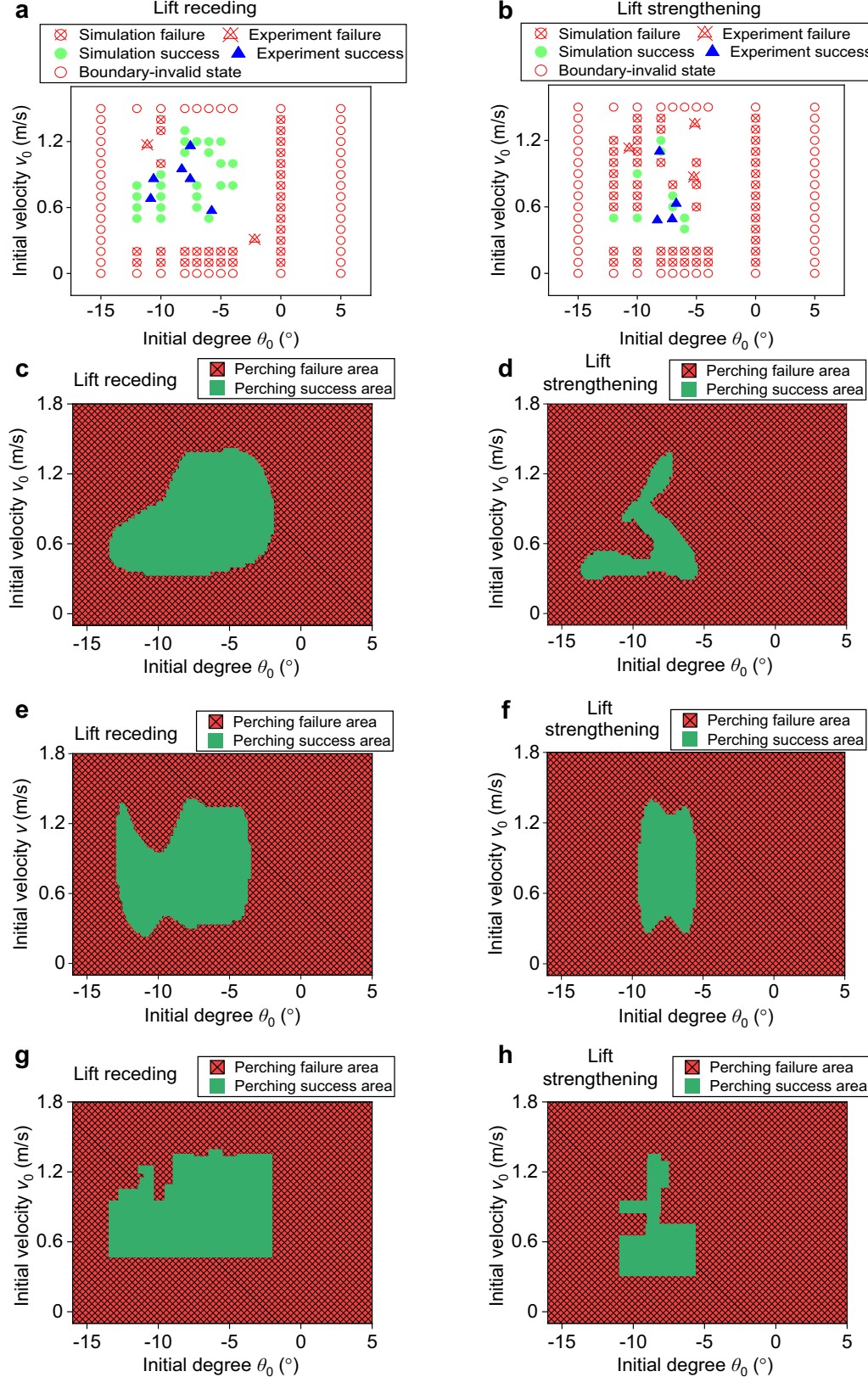

**Fig. 5 | Wall-perching prediction obtained using data-driven model. a** Mixed dataset under lift receding obtained using a knowledge-driven model and experiment. **b** Mixed dataset under lift strengthening obtained using a knowledge-driven model and experiment. **c** The decision boundary obtained by the MLP model under lift receding. **d** The decision boundary obtained by the MLP model under lift strengthening. **e** The decision boundary obtained by the SVM model under lift receding. **f** The decision boundary obtained by the SVM model under lift strengthening. **g** The decision boundary obtained by the RF model under lift receding. **h** The decision boundary obtained by the RF model under lift strengthening.

under two distinct control strategies: lift receding and lift strengthening. Subsequently, based on the data-driven model's predictions of landing success or failure during actual landings, different control strategies can be implemented for the robot. Optimized control is achieved by adjusting the robot's flight parameters.

Although the prediction framework proposed in this paper has only been validated on the flying robot in this design, the closed-loop design analysis mentioned above for the flying robot is also applicable to other wall-perching robots of different structures. That is, through experiments or simulations, obtain a dataset of the relationship between the influencing factors such as speed, posture, and distance of arbitrary robot during its wall-perching and the success of the perching. Then, use the machine learning method employed in this paper to predict the perching success rate under different initial conditions, and further optimize the design of the robot based on the prediction results.

## Discussion

We first discuss the efficiency and accuracy of the three machine learning approaches. A comparative analysis revealed that the MLP training time was only about 5 s, whereas SVM, due to the need to search for optimal parameters, had the longest training time of approximately 180 s−36 times that of MLP. Additionally, by examining the decision boundary results predicted by the three methods, it can be observed that compared to MLP's predictions, SVM and Random Forest exhibited larger errors. Their success regions contained several characteristic points, which corresponded to failure outcomes in both numerical calculations and experimental tests. Conversely, such contradictions were almost absent in the MLP results. Therefore, for the problem addressed in this paper, the MLP method demonstrates relatively higher prediction accuracy.

It is widely acknowledged that traditional methods such as the finite element method or the material point method often require a large amount of computational time to simulate a single landing event. This is primarily due to the nature of the landing event, which presents a typical contact-impact problem. To accurately model this problem, a very small time step is necessary for integrating the high-dimensional dynamic differential equations. Additionally, the iterative computations required to address the contact nonlinearity further contribute to the time-consuming nature of the simulations.

Simulating all possible scenarios with arbitrary initial angles and velocities of a robot using traditional methods is clearly impractical. Similarly, conducting experiments to test every case is also unfeasible. The experimental process is not only time-consuming but also expensive. It involves extensive preliminary preparation and equipment debugging, and there is also the risk of potential damage to the robot during experiments. In conclusion, whether through experimentation or traditional numerical simulation, simulating the entire range of coverage states is challenging. The machine learning-based framework employed in this paper allows for predicting the complete space, encompassing all possible cases, using a reduced number of data samples, thereby enhancing the efficiency of landing research.

Secondly, the primary goal of AI is generally to achieve "learning to simulation". This concept entails training models using data, enabling them to perform simulations more efficiently without complex manual modeling and algorithm design. In this paper, we also employ AI to build a data-driven model. By utilizing this model, we can predict the success or failure of a robot's landing and determine its subsequent behavior. This methodology, known as "learning to simulation", is widely applicable in diverse domains such as computer graphics, robotics, and physics modeling.

However, this paper introduces a concept called "simulation to learning". Unlike the previous work, the data in this study is not solely derived from experiments but also includes simulation data computed by a knowledge-driven model. The combination of this simulation data

and experimental data results in mixed data. The main objective of "simulation to learning" is to obtain mixed data. Subsequently, the mixed data is utilized to establish a data-driven model. Ultimately, this data-driven model enables efficient and accurate simulation of all landing events. Our research validates the effectiveness of the "simulation-to-learning" approach.

Finally, we discuss the potential issues in practical applications. The current study demonstrates successful perching of the robotic platform under low-speed flight conditions. However, extending this capability to high-speed operations presents several challenges: sensor delays in gyroscopes and accelerometers may cause attitude control lag during landing, while limited motor response could destabilize rapid flight-to-perch transitions. Accelerometer noise during impact may also interfere with contact detection, though filtering algorithms could mitigate this effect. Additionally, the control strategy must address motor overheating during prolonged stall conditions through optimized power-thermal management. These challenges highlight important directions for future research in high-speed robotic perching.

This paper focuses on proposing a machine learning-based prediction framework for wall-perching, where the landing experiments aim to collect samples of successful and failed outcomes. Consequently, our experiments solely examine the process from flight state to perched landing, without yet utilizing prediction results to: assess landing success/failure prior to wall contact, guide the robot's decision on whether to initiate landing, or direct it to steer away from the wall to reorient both pitch angle and initial approach velocity for subsequent perching attempts. Implementing this comprehensive closed-loop process−using our predictive framework as the decision-making core −remains future work, which will establish complete control strategies for aerial robotic wall-perching and lay the groundwork for developing feedback regulation algorithms.

In addition, it must be pointed out that the experimental data we used for prediction were measured under relatively ideal laboratory conditions and are not fully applicable to experimental conditions with high interference or smoother walls. Under high interference conditions (such as when the wind is strong), due to the small size of the robot, it will have a significant impact on the stability of flight, thereby affecting the results of landing prediction. However, the grasping structure used by the robot in this study cannot be successfully attached to smooth walls. Therefore, the prediction model proposed in this study is not applicable to the experimental conditions of smooth walls.

## Methods
### Structure and material of robot
The UG software was employed to create a detailed 3D model of the robot (Fig. 1). The robot involves three key systems: flight system, landing system, and attachment system. Ultimately, the Crazyflie 2.1 quadrotor flight platform by Bitcraze was selected. The landing system is composed of two carbon fiber rods, two rubber balls, and two wheels. We have carefully adjusted the thickness and angle of the carbon fiber rods to optimize the energy absorption and conversion capabilities during the landing process. The core components of the attachment system are two spines, which consist of micro-scale curvy metal structures. These spines are connected to curved carbon fiber rods, flexible ropes, a horizontal bar, and a T-shaped bar using revolute joints and holes. The connector parts are 3D printed using resin material. In the flight system, in/out servos and extend/retract servos are employed to control the movement of the two spines.

### Knowledge-driven model and its computational details
The geometric entities are created within the LS-DYNA software (Fig. 2a). The entity consists of two parts: a vertical wall and a robot. The robot itself is divided into six components: quadcopter, servos, carbon fiber frame, connector, spines, and ropes.

Firstly, to capture the transient landing deformation, all parts of the robot are modeled as deformable bodies in terms of the material model. The quadcopter, ropes, and servos are simulated using the 003 PLASTIC isotropic material model to represent plastic behavior. The carbon fiber parts utilize the 054 ENHANCED orthotropic material model to simulate the properties of carbon fiber[23]. On the other hand, the connectors and spine parts are modeled using the 003 PLASTIC isotropic material model to represent resin material. Lastly, the wall is modeled using the 020 RIGID material model to simulate a rigid body.

To facilitate the development of the robot's dynamics equation, we employ the finite element technique to discretize the structure's deformation field and create a mesh for the flexible parts. The resulting discrete model in LS-DYNA is depicted in Fig. 2a. Prior to meshing, the parts are divided into simpler shapes, and techniques such as sweeping and mapping are utilized to generate high-quality meshes. Additionally, based on knowledge-driven modeling, a combination of regular and hexagonal meshes is employed to ensure accurate results (Fig. 2a). The total number of elements in the model amounts to 58232.

Secondly, in the aspect of constraints, all degrees of freedom of the wall are fully constrained, and an automatic surface-to-surface contact algorithm utilizing a penalty function is employed between the robot and the wall. With consideration for computational convenience, neighboring parts are rigidly connected as constraints.

Thirdly, in terms of the loads, the lift of the robot is shown in Fig. 2c and d. The application of the lift exists in two situations: the first is that the lift is always the same as the gravity of the robot. The second is that the lift is the same as the gravity before contacting the surface to simulate normal flight, and then the lift increases to twice the gravity after contact with the wall. The direction of the robot's initial velocity points towards the wall, ranging from 0 m/s to 1.5 m/s, while the robot's initial pitch ranges from 5° to −15°. The prestress of the flexible rope has little effect on the collision process, so it is ignored in the simulation calculation.

Finally, in terms of numerical integration in calculation, the LS-DYNA central difference method is employed to analyze nonlinear explicit dynamics. The size of time step is determined based on the mesh size and set to be $\Delta t = 7.27 \times 10^{-5}$s. By utilizing the LS-DYNA software solver, the system's dynamic response is obtained, which includes the collision contact force, the dynamic stress cloud diagram of the structure, the stress curve for each point, the motion process, and other kinematic parameters.

In summary, the dimensions and material properties of the finite element model used in the numerical simulation closely match those of the actual robot. The constraint conditions between components and loading scenarios align with experimental configurations. Meanwhile, the meshing strategy and mesh density account for grid sensitivity in computational results. Furthermore, for the impact dynamics problem of the landing/perching process, the simulation incorporates contact constraints between the robot and the wall while employing explicit dynamics algorithms. This approach ensures computational accuracy alongside enhanced efficiency. Collective implementation of these measures guarantees strong agreement between numerical simulation and experimental results. To further validate the consistency between simulations and experimental results, the landing outcomes of simulation and experiment are compared for selected data points, as presented in Supplementary Table 2.

### Data-driven model and three machine learning methods

For the establishment of the data-driven model in this paper, three distinct machine learning approaches were employed: Multilayer Perceptron (MLP), Support Vector Machine (SVM), and Random Forest (RF).

MLP is a type of feedforward artificial neural network consisting of an input layer, at least one hidden layer, and an output layer. In an MLP, each layer comprises multiple units. Each unit incorporates a differentiable linear function and an activation function, mapping the input vector of the layer to a scalar output unit. This paper pertains to a binary classification problem with two features: initial velocity and initial angle. The combinations of initial velocity and angle yield a complex non-linear decision boundary (Supplementary Fig. 7). Since MLPs can learn complex interactive relationships between features and thereby fit decision boundaries of intricate shapes, it exhibits high adaptability to the research focus of this paper. Consequently, an MLP model composed of 5 fully connected layers was constructed: Input Layer contains 2 neurons, corresponding to the two features− initial velocity and initial angle. Hidden Layers contain 4 layers with 64, 128, 64, and 32 neurons respectively. The ReLU activation function was applied to these layers to introduce non-linear fitting capability. The Output Layer employs the sigmoid activation function to map the final output to a value representing the probability of a successful landing. After training for 150 epochs, the model has achieved satisfactory predictive performance in determining whether the flying robot successfully lands on the wall for given combinations of initial velocity and angle.

SVM is a supervised learning algorithm mainly used for classification and regression analysis. Its core concept involves finding an optimal hyperplane in a high-dimensional feature space to separate data points of different classes while maximizing the minimum distance between the hyperplane and samples of both classes (i.e., maximizing the margin). For linearly inseparable data, SVM maps original features to a higher-dimensional space via the Kernel Trick to achieve linear separability. In the binary classification problem studied in this paper, the combination of initial velocity and initial angle forms a complex nonlinear decision boundary (Supplementary Fig. 8). The SVM method used in this paper employs the RBF kernel, which utilizes Gaussian functions to project data into an implicit high-dimensional space. This approach effectively captures complex feature interactions, enabling flexible fitting of intricate nonlinear decision boundaries. The specific implementation process is as follows: Use grid search combined with leave-one-out cross-validation to find optimal parameters $c$ (regularization parameter) and $\gamma$ (scaling coefficient of RBF kernel) (Supplementary Table 3). Here, $c$ controls the trade-off between classification errors and margin width−smaller $c$ enhances generalization capability while larger $c$ may cause overfitting; determines the influence range of a single sample−larger values yield more complex decision boundaries. The optimal parameters are determined by maximizing cross-validation accuracy rates, subsequently training the final model.

RF is an ensemble learning method based on decision trees, comprising multiple independent decision trees. When performing classification in this paper using a random forest, an input sample flows to each decision tree, and these trees independently predict the sample's outcome. RF uses the majority vote of classification results from multiple decision trees as the final classification result. This research uses the Gini Index as the feature selection criterion. A smaller Gini Index indicates higher sample purity; therefore, node splitting progresses in the direction of decreasing Gini Index. To achieve better classification performance, hyperparameters such as the maximum depth of decision trees, the maximum number of features considered for splitting a node, and the minimum sample count at leaf nodes and internal nodes are configured. As the number of decision trees critically impacts random forest training results, Leave-One-Out Cross-Validation is employed to determine the optimal number of trees (Supplementary Table 3). After setting these hyperparameters, the original data is input into the random forest for training, yielding the model employed in this paper (Supplementary Fig. 9).

## Data availability

The dataset of robot landing results used to train the machine learning prediction framework has been deposited in the figshare database at

https://doi.org/10.6084/m9.figshare.29636996[24]. Source data are provided with this paper.

## Code availability

The source code is publicly available at https://doi.org/10.5281/zenodo.17473887[25].

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

## Acknowledgements

The authors gratefully acknowledge the support from Robotics and Intelligent Machine Lab founded by Yunian Shen, NJUST. This study was sponsored by the Natural Science Foundation of Jiangsu Province grant (BK20221484 to Y.S.) and the Aeronautical Science Foundation of China (2024M058059002 to Y.S.).

## Author contributions

Y.S. and C.M. contributed to the conceptualization and methodology of the study. Y.S., C.M., Z.Q., and K.L. conducted the investigation. Y.S. provided supervision throughout the research process. Y.S. and C.M. were responsible for writing the original draft of the paper. Y.S., C.M., W.Z., and A.C. contributed to the review and editing of the manuscript. All authors contributed to the writing of the paper.

## Competing interests
The authors declare no competing interests.
