## [Transparent Peer Review file · Nature Communications]

Machine learning-based framework for wall-perching prediction of flying robot

Corresponding Author: Professor Yunian Shen

Version 0:

Reviewer comments:

Reviewer #1

(Remarks to the Author)

The authors of the manuscript entitled "Machine learning-based framework for wall-perching prediction of bioinspired flying robot", present a machine learning-based framework that allows for the wall-perching prediction and structural design of the bio-inspired flying robot with spines. The framework predicts the success or failure of an arbitrary perching event through the combination of simulation and experimental data, which helps robots change their perching behavior. The authors describe a knowledge-based model for calculating various parameters of a robot during perching, and demonstrate five perching behavioral experiments of the robot. This study demonstrates the feasibility of using the combination of simulated and experimental data for robot learning. However, despite the positive impression on the direction and implementation of the study, and the experimental results, I believe there remain various issues to be resolved; some minor and some major (listed below).

Major comments:

- The sentence "For these methods, there are many mature robot designs and applications available [11-12]" uses references that are more than 5 or even 10 years old. Does that mean that no similar studies have appeared in the last 5 years?
- Similarly, the authors cite a number of 10 or even 20+ year old studies in the second paragraph of the introduction; is this representative of the current state of the art of attachment technology? I suggest that the authors investigate research from recent years to enhance the persuasiveness of the statement.
- "Unlike flying animals, which have bimodal locomotion ability, achieving the perching abilities (13) on vertical surfaces is high challenging for flying robots". I doubt whether reference [13] can declare this sentence. In my opinion, it doesn't seem to have anything to do with robots.
- The sentences at the beginning of the third paragraph of the introduction lack literature to support the views. I suggest adding literature to strengthen the credibility of the ideas.
- The authors indicated that "That is, using a robust simulation environment to provide sample data for AI training seems more feasible". The authors' work does not seem to represent this. In other words, the authors use simulated data computed by knowledge-based models, which does not seem to qualify as a robust simulation environment. The statement here is too strong and I suggest that the authors weaken it.
- At the beginning of the results section, the authors introduce the design concept of the robot and its structural features, but the basic implementation of the robot's stable attachment to the vertical wall is not sufficiently clear. It seems to be achieved through the spikes at the end. What is its exact structure like? How is climbing on the wall achieved? I suggest that the authors add details to this section in Figure 1 or in the supplementary material, or cite existing work to demonstrate the effectiveness of this part. I believe this is easy for the author to do.
- The wall in the experiment shown in Figure 3 seems to be inconsistent with the one in the robot experiment shown in Figure 4.

- The authors indicate that this part aims to demonstrate the robot's ability to remain stable under strong crosswinds. However, no experimental validation of this is provided, nor is the intensity of the wind specified. I suggest that the authors clarify the purpose of this experiment or explain its necessity.
- "Figure 4 shows the experimental images of 5 typical landing behaviors, which have been classified based on the numerical evaluation using the knowledge-driven model as shown in Fig. 2A". The authors illustrate that these five typical behaviors are classified by the knowledge-driven model shown in Figure 2A. How exactly is the classification done? The content in Figure 2A seems to be a finite element model; how can it be referred to as a knowledge-driven model (Figure 5A seems more specific)? I suggest that the author optimize this figure.
- In the supplementary video, there seems to be no obvious difference between normal landing, semi-perfect, and perfect landings. The author explains in the manuscript that the distinction between the three behaviors is whether the wheels contact the wall before coming to a stable stop and the number of times the wheels make contact with the wall. However, it is difficult to discern this difference in the video. I suggest that the author add more noticeable details to highlight these differences.
- The authors show that the proposed predictive model is used for adjusting the robot's landing behavior, however, this paper only shows independent experiments for five behaviors. I suggest adding the comprehensive behavioral change experiments shown in Figures 5 F, G, and h to support this statement.
- Overall, I suggest that the author reorganize the logic of the results section, refine necessary explanations and experiments to ensure that readers can smoothly understand the proposed framework.
- This paper uses SVM to predict whether a robot will be able to land successfully. Although this approach achieves a high accuracy rate, have the authors considered other classification methods (e.g., neural network models, etc., which are popular at the moment)?
- For the entire paper, the authors lack a review of the latest relevant literature (within the past three years), which makes it difficult for readers to understand the recent advancements in the field. This is crucial for positioning the research within the current state of the field, especially for publication in top journals like Nature Communications.

Minor comments:

- Figure 1.E repeats "a, b, c" and it is not explained in the caption.
- The meaning of LR and LS needs to be added to the caption of Figure 2 (Although it already appears in the text).
- Figure 4A, D, E, F, and G doesn't show the obvious difference in these behaviors. I suggest choosing representative snapshots.
- In Figure 5, Knowledge-driven model refers to Figure 5 A and Data-driven model and Training refers to Figure 5 D, E? They are both covered with a green background. I suggest to use different colors or areas to distinguish them.
- In the Discussion section, I suggest to add more discussion about the experimental results and methods shown in this paper, and to analyze the reasons for the results, the current problems and the direction of improvement in the future.
- In the bionic landing strategy of robot section of discussion, the authors state a large number of views. However, these views lack relevant references.

Reviewer #2

(Remarks to the Author)

The paper presents a machine learning-based framework for predicting and optimizing the vertical wall adhesion behavior of biomimetic flying robots. By combining knowledge-driven models (finite element dynamic simulations) with data-driven models (support vector machines, SVM), the authors successfully constructed a hybrid sample dataset and achieved the prediction of adhesion success/failure. The research problem holds clear engineering application value, the methodology is innovative, and the experimental results demonstrate high prediction accuracy (96.8%). The paper is well-structured, and the figures and tables are detailed, especially in terms of dynamic response analysis using the knowledge-driven model and hyperplane visualization in the data-driven model. However, some details need to be further supplemented to enhance the rigor and reproducibility of the research.

Suggestions for Improvement:

Method Details Should Be Supplemented:

Data Bias Issue: The hybrid dataset (simulation and experiment) may introduce potential bias. The authors need to explain how consistency between the simulation and experimental data is ensured (e.g., material parameter calibration, boundary condition matching).

Model Comparison and Validation: The paper does not compare with other machine learning methods (e.g., random forests, neural networks). A comparison of the performance of different models should be added to demonstrate the superiority of SVM.

Independent Test Set Validation: Is the 96.8% accuracy based on the training set or an independent test set? The validation process should be clarified to avoid the risk of overfitting.

Experiment and Result Analysis:

Sample Coverage Range: The experiments cover only 114 simulation conditions and limited experimental data. It is recommended to expand the parameter range (e.g., higher initial speeds or extreme angles) to verify the robustness of the model.

Statistical Significance: The selection of parameters c and γ in Figure 6F requires statistical testing (e.g., confidence intervals) to ensure result reproducibility.

Extreme Condition Analysis: The model's performance under extreme conditions, such as surface friction coefficients $\mu < 0.2$ or high dynamic interference, has not been discussed. A limitation analysis should be added to address this.

Figures and Presentation Optimization:

Figure 6 Labeling Unclear: The definition of "Boundary-invalid state" in Figures 6A/B should be clearly explained in the text or figure caption.

Dynamic Process Visualization: It is suggested to add videos or dynamic diagrams (e.g., stress distribution changes during different stages of adhesion) to enhance the intuitiveness of the results.

Terminology Consistency: Some terms (e.g., "lift recede" and "lift strengthen") need to be clearly defined when first introduced.

Literature and Discussion:

Insufficient Literature Citations: Recent related research (e.g., deep reinforcement learning-based dynamic control methods) has not been adequately discussed. A comparison should be added to highlight the uniqueness of this work.

Engineering Implementation Challenges: The paper should discuss potential issues in practical applications (e.g., sensor delays, the effect of motor response speeds on control strategies).

This paper proposes an innovative machine learning framework, providing an efficient solution for predicting the vertical adhesion behavior of biomimetic robots. The methodology design and experimental results have significant academic value. If the above suggestions are addressed and optimized, the rigor and impact of the paper will be further enhanced. It is recommended to accept the paper for publication after revision.

Reviewer #3

(Remarks to the Author)

The authors present a publication on learning based perching maneuvers that is bio-inspired. The work is based on (i) the limitation of current systems able to perch and (ii) the limitations of AI based learning methods which require a lot of training. The authors present a synthetic data set that is used to predict the performance of perching maneuvers and includes an analysis of some animals that can adhere to surfaces. The work is well written and seems technically correct. I have to follow in comments.

1. The bio-inspired narrative is not convincing, in particular it is not clear how the biological inspiration was obtained and how the various animals and animal models are integrated in the design. The link to biology is currently very tentative without a global or transferable analysis of various methods and design functionality that can help in understanding of perching manners. For example the bat perching is a completely different way than the gecko, and combining the two animals in a bio-inspired bionic system is not convincing to be the best method going forward. To address this, I would recommend a more in-depth analysis of many more biological systems and a functional evaluation of various design methods that can enhance perching capability in robots. Currently, the discussion is very high-level in the paper and would need improvement in my opinion. Similarly, I would recommend a more in depth analysis of the state of the art which is very rich with various methods presented by other groups.

2. It is not clear what the transferable innovation the overall system is. In particular, the system is a singular design that has its advantages and disadvantages. For example the impact reduction and damping could be done in various different other ways, using different other designs. Such designs might lead to a completely different behavior and optimum compared to the system that is being built and analyzed in this in this paper. Similarly the FEM analysis is based on certain design decisions, such as using straight rods, which may limit the overall performance potential of the system. In order to address this, I would recommend to include a close loop design analysis where the model, the learning system, and FEM is combined in a design optimization. Without this, the paper remains one particular implementation of the robot, which is nice, but in my opinion lacks a more generalized impact needed for the high impact publications that is targeted here.

While I think that the paper is interesting with a valuable contributions in robotics, I'm not convinced that it is generalized enough and has a broad impact to the field. Therefore a more specialized journal might be a better option for this particular publication.

Version 1:

Reviewer comments:

Reviewer #1

(Remarks to the Author)

The authors have revised their manuscript. Some of the claims in the text have been modified. The authors also clarify the difference between the proposed framework and other research contributions.

Overall, this article has been greatly improved.

However, I believe there remain various issues to be resolved. At the end of the second paragraph of the Introduction, the authors state that “no research to date has established predictive frameworks for perching”. Although this section lists some studies on the dynamic perching process of robots, it does not explain why it is necessary to study the perching prediction framework. This may cause readers to doubt the importance of the framework in the paper.

In addition, the proposed perching prediction framework in this paper appears to be tailored to a specific robot, and it is unclear whether it is generalizable to other types of robots.

The author may need to make some formatting optimizations, such as the size and contrast of the text in the figure.

(Remarks on code availability)

Reviewer #2

(Remarks to the Author)

The modification requirements have been met

(Remarks on code availability)

Version 2:

Reviewer comments:

Reviewer #1

(Remarks to the Author)

All concerns have been resolved. I agree with the final acceptance of this manuscript.

(Remarks on code availability)

REPLY to REVIEWER #1

The authors of the manuscript entitled “Machine learning-based framework for wall-perching prediction of bioinspired flying robot”, present a machine learning-based framework that allows for the wall-perching prediction and structural design of the bio-inspired flying robot with spines. The framework predicts the success or failure of an arbitrary perching event through the combination of simulation and experimental data, which helps robots change their perching behavior. The authors describe a knowledge-based model for calculating various parameters of a robot during perching, and demonstrate five perching behavioral experiments of the robot. This study demonstrates the feasibility of using the combination of simulated and experimental data for robot learning. However, despite the positive impression on the direction and implementation of the study, and the experimental results, I believe there remain various issues to be resolved; some minor and some major (listed below).

Question 1) The sentence “For these methods, there are many mature robot designs and applications available [11-12]” uses references that are more than 5 or even 10 years old. Does that mean that no similar studies have appeared in the last 5 years? Similarly, the authors cite a number of 10 or even 20+ year old studies in the second paragraph of the introduction; is this representative of the current state of the art of attachment technology? I suggest that the authors investigate research from recent years to enhance the persuasiveness of the statement.

Reply: We sincerely appreciate the valuable suggestions made by the reviewers regarding the timeliness of the references in the introduction. In response to this suggestion, we systematically reviewed the recent (2019-2024) research progress and supplemented the relevant contemporary literature at the corresponding position in the introduction. These additions improve the persuasiveness of this paper. Thank you.

Question 2) “Unlike flying animals, which have bimodal locomotion ability, achieving the perching abilities (13) on vertical surfaces is high challenging for flying robots”. I doubt whether reference [13] can declare this sentence. In my opinion, it doesn't seem to have anything to do with robots. The sentences at the beginning of the third paragraph of the introduction lack literature to support

the views. I suggest adding literature to strengthen the credibility of the ideas.

Reply: Thank you for your important comment. Reference [13] was inappropriately cited to support the statement about the challenges of perching for flying robots. We sincerely apologize for this error and have removed it. Additionally, in response to your first question and the editor's feedback, we have thoroughly revised the Introduction section. Regarding the claims in the third paragraph of the Introduction, we have reviewed the relevant literature and added citations to key publications to better support our arguments.

Question 3) The authors indicated that “That is, using a robust simulation environment to provide sample data for AI training seems more feasible”. The authors' work does not seem to represent this. In other words, the authors use simulated data computed by knowledge-based models, which does not seem to qualify as a robust simulation environment. The statement here is too strong and I suggest that the authors weaken it.

Reply: We sincerely appreciate the reviewer's valuable feedback. As the reviewer pointed out, the statement “That is, using a robust simulation environment to provide sample data for AI training seems more feasible” could be potentially misleading for the readers, so we have removed this sentence in the revised manuscript. Thank you.

Question 4) At the beginning of the results section, the authors introduce the design concept of the robot and its structural features, but the basic implementation of the robot's stable attachment to the vertical wall is not sufficiently clear. It seems to be achieved through the spikes at the end. What is its exact structure like? How is climbing on the wall achieved? I suggest that the authors add details to this section in Figure 1 or in the supplementary material, or cite existing work to demonstrate the effectiveness of this part. I believe this is easy for the author to do.

Reply: Many thanks for the reviewer's valuable comments regarding the insufficient elaboration of the wall attachment and climbing mechanisms. In response, we have added a locally magnified diagram (labeled 'e') to Figure 1 to clearly depict the spikes structures on the feet, and supplemented the relevant description of specific structure of the spikes and how to achieve wall climbing in the Supplementary Information.

Question 5) The wall in the experiment shown in Figure 3 seems to be inconsistent with the one in the robot experiment shown in Figure 4. The authors indicate that this part aims to demonstrate the robot's ability to remain stable under strong crosswinds. However, no experimental validation of this is provided, nor is the intensity of the wind specified. I suggest that the authors clarify the purpose of this experiment or explain its necessity.

Reply: The primary objective of this study is to propose a machine learning framework for predicting the dynamic landing behavior of flying robots. As the reviewers pointed out, the experiment presented in Figure 3 does not contribute to illustrating the research objectives and may potentially confuse readers. Therefore, we have removed this section from the revised manuscript. Thanks.

Question 6) "Figure 4 shows the experimental images of 5 typical landing behaviors, which have been classified based on the numerical evaluation using the knowledge-driven model as shown in Fig. 2A". The authors illustrate that these five typical behaviors are classified by the knowledge-driven model shown in Figure 2A. How exactly is the classification done? The content in Figure 2A seems to be a finite element model; how can it be referred to as a knowledge-driven model (Figure 5A seems more specific)? I suggest that the author optimize this figure."

Reply: Thanks for your comment. The classification of landing behaviors is based on the trajectory of the center of mass and the presence of significant post-attachment body oscillations. The detailed classification principles are comprehensively described in the "Numerical evaluation for perching based on knowledge-driven model" section. Figure 2A is indeed a finite element model, which is inherently a knowledge-driven model because it is constructed based on prior physical laws and mathematical principles rather than relying on statistical patterns in data (i.e., data-driven model).

Question 7) In the supplementary video, there seems to be no obvious difference between normal landing, semi-perfect, and perfect landings. The author explains in the manuscript that the distinction between the three behaviors is in whether the wheels contact the wall before coming to a stable stop and the number of times the wheels make contact with the wall. However, it is difficult to discern this difference in the video. I suggest that the author add more noticeable details to

highlight these differences.

Reply: Many thanks for the reviewer's comment. In the updated video, we have added tracer marks to the robot's center of mass. These trajectories can be used to distinguish the differences in landing situations.

Question 8) The authors show that the proposed predictive model is used for adjusting the robot's landing behavior, however, this paper only shows independent experiments for five behaviors. I suggest adding the comprehensive behavioral change experiments shown in Figures 5 F, G, and h to support this statement. Overall, I suggest that the author reorganize the logic of the results section, refine necessary explanations and experiments to ensure that readers can smoothly understand the proposed framework.

Reply: Thanks for your valuable comment. The logic of the results section has been reorganized. And the original Figures 5 F-H were intended to demonstrate an application vision and possibility of the wall-perching prediction framework proposed in this paper. However, achieving a complete demonstration of these behaviors requires addressing technical challenges such as rapid edge computing and real-time collision monitoring, which demand significant time to conquer. These work will be implemented in subsequent studies. We sincerely appreciate the reviewers for pointing out the direction for our future work. This fact has been elaborated on in the revised manuscript. Thank you again for your valuable feedback.

Question 9) This paper uses SVM to predict whether a robot will be able to land successfully. Although this approach achieves a high accuracy rate, have the authors considered other classification methods (e.g., neural network models, etc., which are popular at the moment)?

Reply: According to the reviewer's valuable suggestion, the landing success/failure predictions using Random Forest (RF) and Multi-layer Perceptron (MLP) have been supplemented in the revised manuscript. Support Vector Machine (SVM) code was optimized, and its hyperparameters were improved. A comparison of the three machine learning methods is presented in the revised manuscript.

Question 10) For the entire paper, the authors lack a review of the latest relevant literature (within the past three years), which makes it difficult for readers to understand the recent advancements

in the field. This is crucial for positioning the research within the current state of the field, especially for publication in top journals like Nature Communications.

Reply: Thank you for your valuable comment. According to your suggestion, we researched relevant references on the topic of the manuscript, including papers published in the past five years (2019-2024). To demonstrate the latest progress in the field, several key directly related references have been added in the revised manuscript.

Question 11) Figure 1.E repeats “a, b, c” and it is not explained it in the caption.

Reply: According to the format instruction of nature communications, the repeats “a, b, c” have been deleted. Thank you.

Question 12) The meaning of LR and LS needs to be added to the caption of Figure 2 (Although it already appears in the text).

Reply: Thank you for your comment. We have added the missing meanings of LR and LS to the caption of Figure 2.

Question 13) Figure 4A, D, E, F, and G doesn't show the obvious difference in these behaviors. I suggest choosing representative snapshots.

Reply: Based on the reviewer's feedback, we have reselected representative snapshots to highlight the key characteristics of different landing behaviors. Thank you.

Question 14) In Figure 5, Knowledge-driven model refers to Figure 5 A and Data-driven model and Training refers to Figure 5 D, E? They are both covered with a green background. I suggest to use different colors or areas to distinguish them.

Reply: Thank you for your suggestion. Due to content reorganization, the original Figure 5 has been renumbered as Figure 4 in the revised manuscript. We have color-coded the sections of this figure, with the green area representing the knowledge-driven model, the purple section indicating the landing experiment, and the yellow portion designating the data-driven model.

Question 15) In the Discussion section, I suggest to add more discussion about the experimental results and methods shown in this paper, and to analyze the reasons for the results, the current

problems and the direction of improvement in the future.

Reply: Thanks for your valuable suggestions. Following your suggestions, we have added discussion of the limitations of our experimental approach along with potential improvements for future research in the Discussion section, and have expanded the Supplementary Information with more detailed descriptions of the experimental method.

Question 16) In the bionic landing strategy of robot section of discussion, the authors state a large number of views. However, these views lack relevant references.

Reply: Following the editor's guidance, the paper's focus does not lie in bionics, so we have removed the extensive discussions of bio-inspired landing strategies from the original manuscript. We sincerely appreciate the reviewer's valuable insights.

REPLY to REVIEWER #2

The paper presents a machine learning-based framework for predicting and optimizing the vertical wall adhesion behavior of biomimetic flying robots. By combining knowledge-driven models (finite element dynamic simulations) with data-driven models (support vector machines, SVM), the authors successfully constructed a hybrid sample dataset and achieved the prediction of adhesion success/failure. The research problem holds clear engineering application value, the methodology is innovative, and the experimental results demonstrate high prediction accuracy (96.8%). The paper is well-structured, and the figures and tables are detailed, especially in terms of dynamic response analysis using the knowledge-driven model and hyperplane visualization in the data-driven model. However, some details need to be further supplemented to enhance the rigor and reproducibility of the research.

This paper proposes an innovative machine learning framework, providing an efficient solution for predicting the vertical adhesion behavior of biomimetic robots. The methodology design and experimental results have significant academic value. If the suggestions are addressed and optimized, the rigor and impact of the paper will be further enhanced. It is recommended to accept the paper for publication after revision.

Question 1) Data Bias Issue: The hybrid dataset (simulation and experiment) may introduce potential bias. The authors need to explain how consistency between the simulation and

experimental data is ensured (e.g., material parameter calibration, boundary condition matching).

Model Comparison and Validation: The paper does not compare with other machine learning methods (e.g., random forests, neural networks). A comparison of the performance of different models should be added to demonstrate the superiority of SVM.

Independent Test Set Validation: Is the 96.8% accuracy based on the training set or an independent test set? The validation process should be clarified to avoid the risk of overfitting.

Reply: In the revised manuscript, the Methods section elaborates on the numerical simulation methodology for robotic landing, with detailed justifications for the consistency between simulations and experimental results from perspectives of model setup, material properties, boundary conditions, and loading configurations. Furthermore, a comparative study incorporating Support Vector Machine (SVM), Random Forest (RF), and Multilayer Perceptron (MLP) was newly conducted. The results demonstrate approximately comparable prediction results among the three methods, yet MLP exhibits better performance. Thus, we adopted MLP as the optimal model for predicting robotic landing dynamics, and corresponding revisions have been implemented in the manuscript. The accuracy metrics in this study are derived from leave-one-out cross-validation (LOOCV). In each training iteration, one sample is reserved as the test set while all others constitute the training set, cycling through all samples. This approach rigorously ensures independence across iterations and effective mitigation of overfitting risks. The relevant information has been added to the revised manuscript. Thanks for your valuable comment.

Question 2) Experiment and Result Analysis: Sample Coverage Range: The experiments cover only 114 simulation conditions and limited experimental data. It is recommended to expand the parameter range (e.g., higher initial speeds or extreme angles) to verify the robustness of the model.

Statistical Significance: The selection of parameters c and γ in Figure 6F requires statistical testing (e.g., confidence intervals) to ensure result reproducibility.

Extreme Condition Analysis: The model's performance under extreme conditions, such as surface friction coefficients $\mu < 0.2$ or high dynamic interference, has not been discussed. A limitation analysis should be added to address this.

Reply: Following your recommendations, we expanded the parameter range to include higher

initial horizontal velocities and approach angles. Subsequent prediction analyses reveal that incorporating these additional data points does not alter existing outcomes, as all new entries correspond to perching failure cases. Besides, to ensure the stability and reproducibility of our results, leave-one-out cross-validation (LOOCV) was employed throughout this study. This validation method rigorously trains the model under specific hyperparameters across all possible data partitions, ensuring every sample in the dataset participates in the validation process. Consequently, the selected hyperparameters attain robust confidence. Our experiment was conducted under relatively ideal conditions, so our predicted results may not apply in scenarios such as high interference or on smooth surfaces. A more detailed discussion has been added to the discussion section in the revised manuscript. Many thanks for your helpful comments.

Question 3) Literature and Discussion: Insufficient Literature Citations: Recent related research (e.g., deep reinforcement learning-based dynamic control methods) has not been adequately discussed. A comparison should be added to highlight the uniqueness of this work.

Reply: In the revised manuscript, we have expanded the Introduction to incorporate a discussion of recent advances in relevant studies over the past several years. Thank you.

Question 4) Engineering Implementation Challenges: The paper should discuss potential issues in practical applications (e.g., sensor delays, the effect of motor response speeds on control strategies).

Reply: We appreciate your feedback. The discussion of potential implementation challenges has been expanded in the Discussion section of the revised manuscript.

REPLY to REVIEWER #3

The authors present a publication on learning based perching maneuvers that is bio-inspired. The work is based on (i) the limitation of current systems able to perch and (ii) the limitations of AI based learning methods which require a lot of training. The authors present a synthetic data set that is used to predict the performance of perching maneuvers and includes an analysis of some animals that can adhere to surfaces. The work is well written and seems technically correct. I have to follow in comments.

Question 1) The bio-inspired narrative is not convincing, in particular it is not clear how the biological inspiration was obtained and how the various animals and animal models are integrated in the design. The link to biology is currently very tentative without a global or transferable analysis of various methods and design functionality that can help in understanding of perching manners. For example the bat perching is a completely different way than the gecko, and combining the two animals in a bio-inspired bionic system is not convincing to be the best method going forward. To address this, I would recommend a more in-depth analysis of many more biological systems and a functional evaluation of various design methods that can enhance perching capability in robots. Currently, the discussion is very high-level in the paper and would need improvement in my opinion. Similarly, I would recommend a more in depth analysis of the state of the art which is very rich with various methods presented by other groups.

Reply: Thank you for your comment. Our articles mainly focus on the research of the landing prediction system for flying robots on the wall surface. To avoid misleading readers to overly focus on the biological inspiration, and in accordance with the strong suggestion from the editor "strongly tone down the bio-inspiration", we have decided to delete the content related to biological inspiration and have revised the article to focus on the introduction and analysis of the prediction framework. Our future work will conduct in-depth investigations into bionics-inspired mechanisms, with detailed exploration planned for subsequent research. We sincerely appreciate the reviewer for highlighting this valuable direction.

Question 2) It is not clear what the transferable innovation the overall system is. In particular, the system is a singular design that has its advantages and disadvantages. For example the impact reduction and damping could be done in various different other ways, using different other designs. Such designs might lead to a completely different behavior and optimum compared to the system that is being built and analyzed in this in this paper. Similarly the FEM analysis is based on certain design decisions, such as using straight rods, which may limit the overall performance potential of the system. In order to address this, I would recommend to include a close loop design analysis where the model, the learning system, and FEM is combined in a design optimization. Without this, the paper remains one particular implementation of the robot, which is nice, but in my opinion

lacks a more generalized impact needed for the high impact publications that is targeted here.

Reply: Thank you for your highly valuable comment. Traditional FEM-based design methods can only evaluate landing performance under limited attitude parameters for specific robot configurations. In contrast, the proposed machine learning framework enables comprehensive perching evaluation across nearly all attitude parameters. Consequently, this facilitates rapid assessment of whether structural optimizations are required. In fact, we have already employed the proposed machine learning framework for closed-loop optimization during structural and attitude parameter design.

The initial manuscript focused on introducing the framework itself, inadvertently ignoring its pivotal role in the design process. We sincerely appreciate the reviewer's reminder on this point. Following your suggestion, the closed-loop design analysis procedure has been added to the revised manuscript. Through this analysis, we optimized the structure and attitude parameters of our flying robot, selecting a better configuration from multiple candidates. The incorporation of this closed-loop design analysis further enhances the transferability of our machine learning framework in the field of wall-landing robot design, amplifying its broader impact. Thank you!

REPLY to REVIEWER #1

Question 1) The authors have revised their manuscript. Some of the claims in the text have been modified. The authors also clarify the difference between the proposed framework and other research contributions.

Overall, this article has been greatly improved.

However, I believe there remain various issues to be resolved. At the end of the second paragraph of the Introduction, the authors state that “no research to date has established predictive frameworks for perching”. Although this section lists some studies on the dynamic perching process of robots, it does not explain why it is necessary to study the perching prediction framework. This may cause readers to doubt the importance of the framework in the paper.

Reply: We fully agree with your suggestions and have revised the manuscript accordingly. In particular, the discussion at the end of the second paragraph in the Introduction has been expanded to better highlight the significance of our research. This enhancement is intended to improve readers' understanding of the value of the proposed framework. Thank you for your valuable comment.

Question 2) In addition, the proposed perching prediction framework in this paper appears to be tailored to a specific robot, and it is unclear whether it is generalizable to other types of robots.

Reply: Thank you for your valuable feedback. Your suggestion is instrumental in improving the generalizability of the framework proposed in this paper. Accordingly, in the "Closed-loop Design Analysis" section of the Results, we have incorporated a discussion regarding the generalizability of the machine learning-based closed-loop design framework introduced in this study, along with an outline of its design process.

Question 3) The author may need to make some formatting optimizations, such as the size and contrast of the text in the figure.

Reply: The formatting of the entire manuscript has been carefully reviewed. We have also optimized the text sizes in Figures 1 and 2. Thank you once again for your valuable suggestion.

REPLY to REVIEWER #2

Question 1) The modification requirements have been met.

Reply: Thank you very much for your recognition of our research.

REPLY to REVIEWER #1

Question 1) All concerns have been resolved. I agree with the final acceptance of this manuscript.

Reply: Thank you very much for your positive feedback and for accepting our manuscript.